# Candidate Causal Variants at the 8p12 Breast Cancer Risk Locus Regulate *DUSP4*

**DOI:** 10.3390/cancers12010170

**Published:** 2020-01-10

**Authors:** Dylan M. Glubb, Wei Shi, Jonathan Beesley, Laura Fachal, Jayne-Louise Pritchard, Karen McCue, Daniel R. Barnes, Antonis C. Antoniou, Alison M. Dunning, Douglas F. Easton, Georgia Chenevix-Trench

**Affiliations:** 1Department of Genetics and Computational Biology, QIMR Berghofer Medical Research Institute, Brisbane QLD 4006, Australia; Wei.Shi@qimrberghofer.edu.au (W.S.); Jonathan.Beesley@qimrberghofer.edu.au (J.B.); jaynelouise.pritchard@gmail.com (J.-L.P.); Karen.McCue@qimrberghofer.edu.au (K.M.); Georgia.Trench@qimrberghofer.edu.au (G.C.-T.); 2Centre for Cancer Genetic Epidemiology, Department of Oncology, University of Cambridge, Cambridge CB1 8RN, UK; lf9@sanger.ac.uk (L.F.); amd24@medschl.cam.ac.uk (A.M.D.); dfe20@medschl.cam.ac.uk (D.F.E.); 3Centre for Cancer Genetic Epidemiology, Department of Public Health and Primary Care, University of Cambridge, Cambridge CB1 8RN, UK; drb54@medschl.cam.ac.uk (D.R.B.); aca20@medschl.cam.ac.uk (A.C.A.)

**Keywords:** breast cancer risk, GWAS, candidate causal variant, chromatin conformation capture, reporter gene activity, enhancer, promoter

## Abstract

Genome-wide association studies have revealed a locus at 8p12 that is associated with breast cancer risk. Fine-mapping of this locus identified 16 candidate causal variants (CCVs). However, as these variants are intergenic, their function is unclear. To map chromatin looping from this risk locus to a previously identified candidate target gene, *DUSP4*, we performed chromatin conformation capture analyses in normal and tumoural breast cell lines. We identified putative regulatory elements, containing CCVs, which looped to the *DUSP4* promoter region. Using reporter gene assays, we found that the risk allele of CCV rs7461885 reduced the activity of a *DUSP4* enhancer element, consistent with the function of *DUSP4* as a tumour suppressor gene. Furthermore, the risk allele of CCV rs12155535, located in another *DUSP4* enhancer element, was negatively correlated with looping of this element to the *DUSP4* promoter region, suggesting that this allele would be associated with reduced expression. These findings provide the first evidence that CCV risk alleles downregulate *DUSP4* expression, suggesting that this gene is a regulatory target of the 8p12 breast cancer risk locus.

## 1. Introduction

Genome-wide association studies by the Breast Cancer Association Consortium (BCAC) and Consortium of Investigators of Modifiers of BRCA1/2 (CIMBA) have previously found genetic variants at 8p12 associated with breast cancer risk [1,2,3,4]. In the largest study, the most significant association was for the rarer allele (A) of single nucleotide polymorphism rs9693444 with a per-allele odds ratio (OR) = 1.06 (95% CI 1.04–1.08; *p* = 2 × 10^−21^) [1]. Subsequent fine-mapping analysis of BCAC and CIMBA data at this locus revealed 16 candidate causal variants (CCVs; Appendix A), including rs9693444, which could not be statistically separated from each other using a likelihood ratio criterion of 100:1 [5]. The CCVs are located in an intergenic region ≥300 kb centromerically from *DUSP4* and ≥285 kb telomerically from the microRNA *MIR3148*.

A major obstacle in the follow-up of genome-wide association study (GWAS) loci is the identification of target genes. Indeed, given the intergenic location of the CCVs at 8p12, it is unclear what function the risk variation may have. To identify breast cancer GWAS target genes, we previously developed a pipeline called INQUISIT (integrated expression quantitative trait and in silico prediction of GWAS targets). INQUISIT incorporates functional genomic data from breast cell lines and tissues, providing a score to prioritise identified target genes [3,5]. In our latest fine-mapping study, INQUISIT revealed seven target genes at the 8p12 breast cancer risk locus, of which only *DUSP4* was a high confidence target [5]. This prediction was partly based on evidence of long-range regulatory chromatin interactions between CCVs and the *DUSP4*, promoter which included RNA polymerase II chromatin interaction analysis by paired-end tag sequencing (ChIA-PET) data in a breast cancer cell line [6], Hi-C chromatin-interaction and bioinformatic analysis identifying an enhancer coincident with CCVs that interacts with the *DUSP4* promoter [7], and capture Hi-C data from breast cancer cell lines [8].

The chromatin interaction data identified through INQUISIT suggest that CCVs could regulate *DUSP4* expression through enhancer–promoter interactions. This may be a common mechanism through which GWAS variants act (reviewed in [9]). Indeed, at a number of breast cancer GWAS risk loci, we have previously shown that CCVs in regulatory elements loop to promoters and regulate their activity [3,10,11]. In the current study, we mapped long-range chromatin interactions between the 8p12 risk locus and the promoter region of *DUSP4*. We provide evidence that a CCV, coincident with a putative regulatory element (PRE) that interacts with *DUSP4*, can regulate *DUSP4* promoter activity. Furthermore, we found a CCV in a different PRE that was associated with allele-specific chromatin looping to the *DUSP4* promoter region. These data suggest strongly that *DUSP4* is a target gene of breast cancer risk variation at the 8p12 locus.

## 2. Results

### 2.1. Chromatin Conformation Capture (3C) Analyses Confirmed That the 8p12 Risk Locus Physically Interacted with the Promoter Region of DUSP4

To map chromatin interactions between specific regions of the 8p12 risk locus (defined by EcoRI restriction fragments; Figure 1) and the *DUSP4* promoter in breast cells, chromatin conformation capture (3C) analyses were conducted in breast cancer (MCF-7 and T-47D) and normal mammary epithelial (Bre-80) cell lines. Some level of interaction with the *DUSP4* promoter was observed in all three cell lines across the 3C restriction fragments at the risk locus (Figure 1). Notably, this broad level of activity is not often observed in 3C analyses. For example, the 2q35 and 5q11.2 breast cancer risk loci are considerably closer (≤100 kb) to corresponding promoter targets than 8p12 CCVs are to the *DUSP4* promoter region, yet many restriction fragments at these loci have no observable promoter interaction [10,11]. At the 8p12 locus, three 3C restriction fragments (#10, #11 and #12) with Hi-C and/or ChIA-PET evidence of chromatin looping to the *DUSP4* promoter demonstrated peaks of interaction in the breast cell lines (Figure 1), highlighting two fragments (#11 and #12) which contain CCVs (two and three, respectively) for further analysis.

### 2.2. CCV rs7461885 Reduced the Enhancer Activity of PRE1 on the DUSP4 Promoter

From analysis of the 3C, ChIA-PET, and epigenetic data at the 8p12 locus (Figure 1), CCVs in two putative regulatory elements (PRE1-2) were prioritized for reporter gene analyses to determine their effects on DUSP4 promoter activity (Figure 2). PRE1, containing three CCVs, significantly enhanced DUSP4 promoter activity by 2.3- and 2.5-fold in Bre-80 and MCF-7 cells, respectively (Figure 2a,b). In these two cell lines, the risk-associated allele of rs7461885, whether as a single variant or in a haplotype with the risk alleles of two other CCVs, significantly reduced the PRE1 enhancer activity by 1.3–1.5 fold (Figure 2a,b). Analysis of rs7461885 using HaploReg [12] did not indicate any evidence for transcription factor motifs or binding; however, this variant is located in active histone marks (H3K4Me1, H3K4Me2, and H3K27Ac) in normal mammary epithelial cells (Figure 1). PRE2, containing two CCVs, significantly enhanced DUSP4 promoter activity by 2.5- and 2.7-fold in MCF-7 and Bre-80 cells, respectively, but neither CCV affected this enhancer activity, nor did the haplotype containing both risk alleles (Figure 2c,d). Reporter assays were also performed in T-47D cells, but neither PREs nor CCVs significantly affected *DUSP4* promoter activity (Appendix A).

### 2.3. CCV rs12155535 (PRE2) Was Associated with Allele-Specific Looping to the DUSP4 Promoter Region

Of the three breast cell lines used in the 3C assay, only one (Bre-80) was heterozygous for CCVs and thus amenable for allele-specific 3C analysis. After assessing CCVs in the PREs and their proximity to the 3C restriction sites, we found that only PRE2 could be analysed for allele-specific looping using PCR-based Sanger sequencing. Using Bre-80 3C libraries, we found that there was a reduction of the risk (C) allele of CCV rs12155535 in PRE2 interactions, compared to the Bre-80 genomic DNA control (Figure 3). This finding indicates that the risk allele was negatively correlated with the PRE2 enhancer-*DUSP4* promoter region interaction, suggesting that risk allele may be associated with diminished *DUSP4* expression through a reduction in enhancer looping to the *DUSP4* promoter region.

## 3. Discussion

In this study, we used genetic approaches to identify functional CCVs and regulatory targets at the 8p12 breast cancer risk locus. Firstly, we performed 3C in normal and breast cancer cell lines to map chromatin interactions between PREs, containing CCVs, and the *DUSP4* promoter region. Although we cannot rule out interactions with other genes at this locus, these findings were consistent with available chromatin interaction analyses, and no other high-confidence targets were predicted by the INQUISIT pipeline in this region [5].

Secondly, we prioritised two PREs with evidence of interaction with the *DUSP4* promoter region (i.e., PRE1-2) for reporter gene analyses. These assays demonstrated that PRE1 and PRE2 acted as enhancers of *DUSP4* promoter activity in both normal and tumoural breast cell lines. PRE1 and PRE2 are located in areas of active chromatin, as indicated by the presence of transcription factor binding and strong histone peaks characteristic of enhancers (Figure 1). CCVs in PRE1 and PRE2 were tested for effects on the *DUSP4* promoter, and the risk allele of CCV rs7461885 was found to repress the *DUSP4* enhancer activity of PRE1. It is possible that other CCVs at the 8p12 locus, which were not assessed in the reporter gene assays, may also regulate *DUSP4*; however, the available epigenetic data suggested these variants were less likely to be functional.

Finally, we performed an allele-specific 3C analysis and found that the risk allele of CCV rs12155535 was present less often in PRE2-*DUSP4* promoter region interactions in comparison with the protective allele. As the reporter gene assays had demonstrated that PRE2 acted as a *DUSP4* enhancer, the association of the rs12155535 risk allele with reduced PRE2-*DUSP4* interaction frequency was consistent with the negative effect of the risk allele of rs7461885 on *DUSP4* promoter activity.

The 3C and reporter gene findings indicate that *DUSP4* is a likely target gene and is downregulated by risk associated variation at the 8p12 locus. A limitation of the reporter gene analysis is that because it is plasmid-based it does not model long-range interaction regulatory effects that occur in the human genome. Nonetheless, enhancers are understood to maintain their functions in reporter plasmids, despite the artificial genetic environment, and reporter assays are considered the “gold standard” for assessing enhancer activity [13]. Indeed, data suggest that distance is not essential for regulatory interaction in a plasmid-based assay—Tewhey et al. estimated a positive predictive value of up to 68% when validating expression quantitative trait loci (eQTLs) using reporter plasmids that incorporate only 150 bp of enhancer sequence and a minimal promoter [14].

Expression quantitative trait loci data from mammary tissue (*n* = 396) in the Genotype Tissue Expression database (version 8; https://gtexportal.org/home/ accessed on 9 October 2019) and tumour tissue (*n* = 799) in The Cancer Genome Atlas eQTL browser (https://albertlab.shinyapps.io/tcga_eqtl/ accessed on 9 October 2019) provided no evidence that CCVs are associated with *DUSP4* expression in normal or tumour breast tissue. A lack of CCV eQTLs at cancer risk GWAS loci is a common finding [10,11,15,16] and could be due to several issues: (i) the available eQTL studies may not have enough statistical power to detect modest effects (as observed in the reporter gene assay) on *DUSP4* expression, for example, eQTLs were found at a third of breast cancer GWAS risk loci only after combining data from four relevant eQTL studies (*n* = 2820) [17]; (ii) the tissues used, especially tumour tissue, were heterogeneous, containing multiple cell types, and thus may have masked CCV effects on *DUSP4* expression that occur in specific cell types; and (iii) CCVs could have development or context-specific effects on *DUSP4* expression that were not captured by the available studies.

*DUSP4* encodes a nuclear dual specificity protein phosphatase that inactivates ERK, JNK, and p38 mitogen-activated protein kinases [18] and may also play a regulatory role in gene expression through chromatin binding [19]. *DUSP4* is frequently deleted in breast tumours [20,21] and cancer cell lines [22]. *DUSP4* knockdown enhances the formation of breast cancer stem cells [19,23] and increases the invasive ability of estrogen receptor-positive breast cancer stem cells [23]. In ER-negative breast cancer cells, *DUSP4* knockdown increases the formation of mammospheres and the expression of cancer stem cell promoting cytokines [22]. Thus, the literature indicates that DUSP4 protein acts as a tumour suppressor in breast cancer, which is compatible with our finding of its negative regulation by a breast cancer risk allele.

## 4. Materials and Methods

### 4.1. Cell Culture

MCF-7 and T-47D cell lines were purchased from ATCC (#HTB22 and #HTB-133, respectively), and Bre-80 was kindly provided by Roger Reddel (CMRI, Sydney, Australia). Cell lines were stored in liquid nitrogen vapour phase with *Mycoplasma* testing and short tandem repeat profiling performed for cell authentication prior to storage. The immortalised mammary epithelial Bre-80 cell line was cultured as previously described [24]. The breast cancer cell lines MCF-7 and T-47D were cultured in RPMI 1640 medium supplemented with fetal calf serum, penicillin/streptomycin, and 10 μg/ml insulin. All cell lines were cultured at 37 °C in a humidified 5% CO_2_ atmosphere.

### 4.2. C Analysis

3C libraries were generated from MCF-7, T-47D, and Bre-80 cells. Briefly, cells were grown in 100 mm plates and fixed with 1% formaldehyde after reaching ≈80% confluency. Formaldehyde was quenched with ice-cold 0.125 M glycine in phosphate buffered saline and cells collected by scraping and centrifugation. After washing with PBS, cells were incubated in ice-cold cell lysis buffer (10 mM Tris pH 7.5, 10 mM NaCl, 0.2% IGEPAL and cOmplete protease inhibitors (Roche)) for 30 min on ice, followed by 10 strokes of a Dounce homogenizer. Cell nuclei were collected by centrifugation and incubated overnight at 37 °C with 1500 U of EcoRI in New England Biolabs restriction buffer with 0.3% sodium dodecyl sulfate and 2% Triton-X 100. Restriction enzyme was heat-inactivated at 65 °C for 20 min and ligation was performed in 8 mL of ligation buffer (1% Triton X-100, 1.15× NEB ligation buffer, 0.1 mg/mL bovine serum albumin and 1 mM ATP). 3C libraries were then phenol/chloroform extracted and precipitated with ethanol.

3C interactions were quantified by qPCR with primers designed with the EcoRI restriction fragments spanning the 8p12 risk locus (Appendix A). qPCR was performed using a RotorGene 6000 with a reaction mix containing MyTaq HS DNA polymerase and the addition of 5 mM Syto9. Thermal cycling was performed with an annealing temperature of 66 °C (20 s) and extension at 72 °C (30 s). Three independent 3C libraries were analysed by qPCR with each experiment quantified in duplicate. Two bacterial artificial clones (RP11-56L5 and RP11-833O14: BACPAC Resource Center) encompassing the *DUSP4* promoter region and the 8p12 breast cancer risk locus were used to create a library of ligation products to determine 3C primer efficiencies for normalization.

### 4.3. Reporter Gene Vector Construction

A *DUSP4* promoter luciferase reporter construct was generated by inserting 1650 bp of PCR-amplified DNA, containing the *DUSP4* transcription start site (chr8:29,207,878-29,209,527; GRCh37), into the KpnI and HindIII sites of a pGL3-Basic construct. The pGL3-Basic construct had been engineered to include AgeI and SbfI sites downstream of the luciferase gene. A 2464 bp PRE1 region (chr88:29,527,323-29,530,033) was PCR generated using primers engineered with BamHI and SalI sites for insertion downstream of the Firefly luciferase gene. A 980 bp PRE2 region (chr8:29522728-29523707) was synthesised with terminal AgeI/SalI sites (Integrated DNA Technologies, Singapore, Singapore) for cloning into the *DUSP4* promoter construct. For PRE1, the allelic variants of CCVs were introduced by overlap extension PCR. For PRE2, constructs containing allelic variants were also synthesised by Integrated DNA Technologies and cloned as above. All constructs were Sanger sequenced (QIMR Berghofer sequencing facility) to confirm variant incorporation. All PCR and sequencing primers are listed in Appendix A.

### 4.4. Reporter Gene Analysis

Bre-80, MCF-7, and T-47D cells were transfected with equimolar amounts of luciferase reporter constructs and 50 ng of the Renilla luciferase pRL-SV40 construct with Lipofectamine 2000. The total amount of transfected DNA was kept constant at 600 ng for each construct by adding pUC19 as a carrier plasmid. Luciferase activity was measured 24 h post-transfection by the Dual-Glo Luciferase Assay System. To correct for variation in transfection efficiency or cell lysate preparation, Firefly luciferase activity was normalised to that of the Renilla luciferase. All *DUSP4* promoter constructs had great activity compared with the negative control (empty pGL3-Basic construct). Data were log-transformed and statistical significance was assessed by two-way ANOVA, followed by Dunnett’s multiple comparisons test in GraphPad Prism (version 7.02, GraphPad Software, San Diego, CA, USA).

### 4.5. Allele-Specific 3C Analysis

DNA was amplified by PCR of two independent Bre-80 3C libraries using the *DUSP4* promoter bait primer and a reverse primer specific to the PRE2 fragment containing rs12155535 (Appendix A). The region containing rs12155535 was also amplified from Bre-80 genomic DNA as a control. Sanger sequencing (QIMR Berghofer sequencing facility) was performed to determine the alleles present in the interacting PRE2 fragment.

## 5. Conclusions

We found the first evidence that breast cancer risk variation at the 8p12 locus downregulates *DUSP4* promoter activity and is negatively correlated with long-range chromatin looping interactions with the *DUSP4* promoter region. These findings are consistent with the role of *DUSP4* as a tumour suppressor in breast cancer and suggest that the effects of CCVs on breast cancer risk may be mediated by a reduction in *DUSP4* expression.

## Figures and Tables

**Figure 1 cancers-12-00170-f001:**
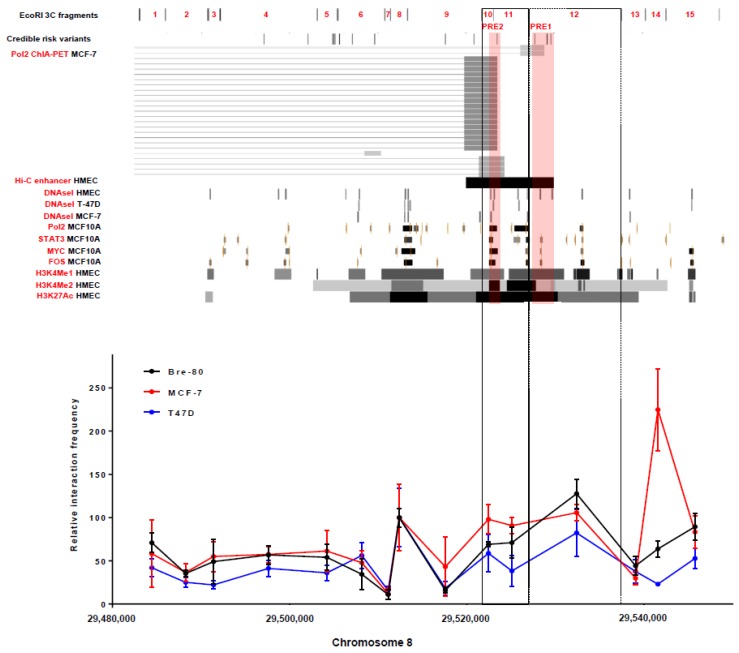
Candidate risk variants (CCVs) were located in putative regulatory elements (PREs) that interact with the *DUSP4* promoter. The figure shows the 8p12 breast cancer risk locus annotated with breast cell functional genomic data and 3C analyses of interactions between EcoRI fragments and the *DUSP4* promoter region (located ≥300 kb upstream) in Bre-80, MCF-7, and T-47D cells. Functional genomic data from normal mammary epithelial cells (HMEC and MCF-10A) and breast cancer cell lines (MCF-7 and T-47D) in this figure include Hi-C analysis [7] and ENCODE data accessed from the Univeristy of California Santa Cruz Genome Browser: RNA polymerase II chromatin interaction analysis by paired-end tag sequencing (ChIA-PET), epigenetic marks characteristic of enhancers (DNaseI hypersensitivity sites and H3K4Me1, H3K4Me2, and H3K27Ac histone modifications), and transcription factor binding (Pol2, STAT3, MYC, and FOS). Notably, all ChIA-PET anchors and the predicted Hi-C enhancer looped to the *DUSP4* promoter. For the 3C analyses, interaction frequencies were normalised to those of fragment 8, a common peak of interaction in all three cell lines. Interaction frequencies from three independent biological replicates are shown (error bars represent standard error of the mean). Fragments demonstrating interaction peaks and encompassing PREs with CCVs are bounded by boxes. The PRE regions cloned for reporter gene analysis are highlighted in pink.

**Figure 2 cancers-12-00170-f002:**
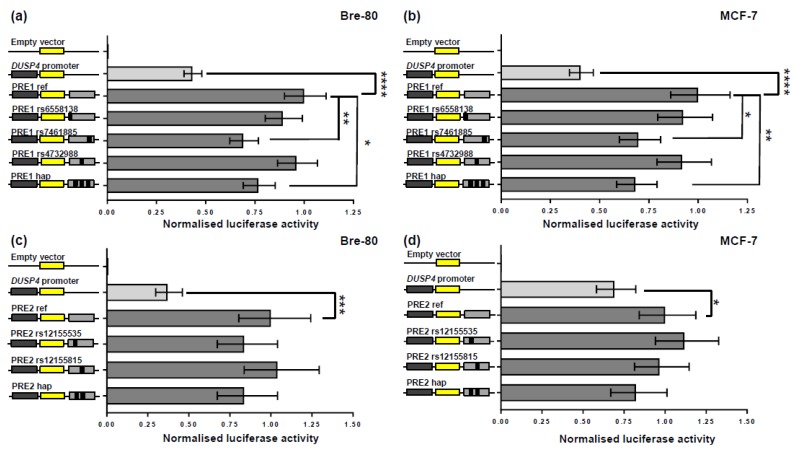
PREs enhanced *DUSP4* promoter activity in luciferase reporter assays and the risk allele of CCV rs7461885 reduced putative regulatory element 1 (PRE1) enhancer activity. PRE1 and PRE2 reference regions, containing protective allelic variants of CCVs, were cloned downstream of a *DUSP4* promoter luciferase construct for the creation of reference (ref) constructs. Risk allelic variants of CCVs were engineered into the constructs and were designated by the rs ID of the corresponding variant. Constructs containing risk allele haplotypes (hap) were also generated. Cells were transiently transfected with each of these constructs and assayed for luciferase activity after 24 h. Panels show back-transformed data for (**a**) PRE1 activity in Bre-80 cells, (**b**) PRE1 activity in MCF-7 cells, (**c**) PRE2 activity in Bre-80 cells, and (**d**) PRE2 activity in MCF-7 cells. Error bars denote 95% confidence intervals of experiments performed in triplicate. *p*-values were determined two-way ANOVA followed by Dunnett’s multiple comparisons test (* *p* < 0.05, ** *p* < 0.01, *** *p* < 0.001 and **** *p* < 0.0001).

**Figure 3 cancers-12-00170-f003:**
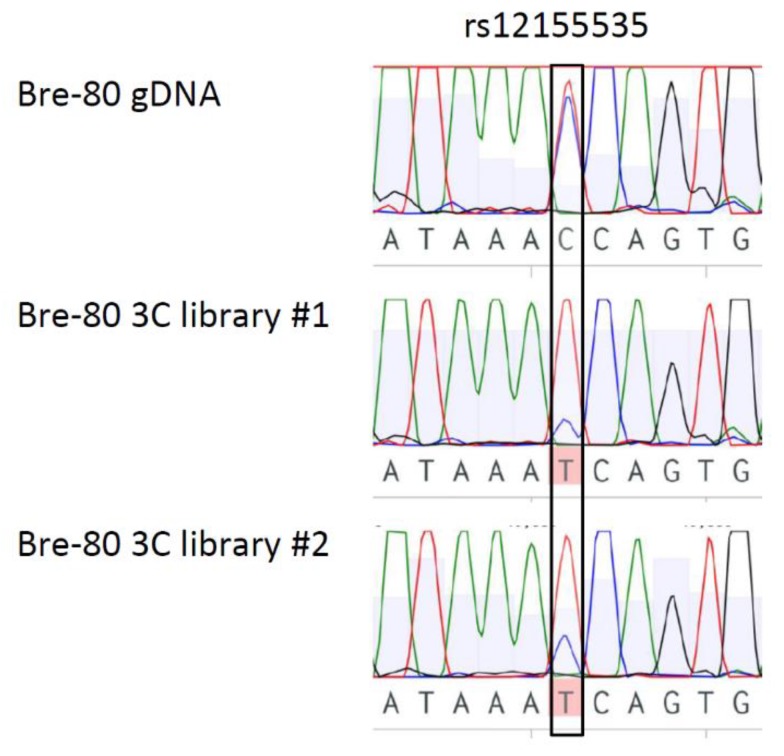
Allele-specific interaction between PRE2 and the *DUSP4* promoter region. Bre-80 cells are heterozygous for the rs12155535 CCV in PRE2. Allele-specific analysis using PCR-based Sanger sequencing of two independent 3C libraries indicated that sequence containing the protective (T) allele of rs12155535 preferentially interacted with the *DUSP4* promoter region.

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
