# Peer review of "Candidate Causal Variants at the 8p12 Breast Cancer Risk Locus Regulate DUSP4"

_cancers, 2020, doi:10.3390/cancers12010170_

Round 1

Reviewer 1 Report

The authors performed chromatin conformation capture analyses in normal and tumoral breast cell lines to map chromatin looping from risk locus to a previously identified candidate target gene, DUSP4. The authors identified putative regulatory elements, containing CCVs, that loop to the DUSP4 promoter region. These findings provide evidence that CCV risk alleles downregulate DUSP4 expression, suggesting that this gene is a regulatory target of the 8p12 breast cancer risk locus. While this is a clear and well-written manuscript, there is insufficient information on the results of previous studies for the reader to follow the rationale and procedures of this study. The methods are usually appropriate, but some details should be provided and the reasons for using this particular method should be provided.  Specific comments follow

There is an error in the title of 4.2

The title of 4.3 is the same as the title of 4.4

Author Response

While this is a clear and well-written manuscript, there is insufficient information on the results of previous studies for the reader to follow the rationale and procedures of this study.

We are unsure as to what specific information the reviewer requires from the previous studies. However, as stated in the Introduction, a locus at 8p12 had been previously associated with breast cancer risk through genome-wide association studies and, as a result of data from multiple analyses that showed long-range chromatin interactions with the DUSP4 promoter region, DUSP4 was identified by INQUISIT as a high confidence target at this locus. To clarify the INQUISIT approach, we have added a further sentence to the Introduction (see lines 45-6).

The methods are usually appropriate, but some details should be provided and the reasons for using this particular method should be provided. 

We have provided more detail to the 3C method in the 4.2 of the Materials and Methods section. We have clarified the rationale for using this method in lines 66-7.

There is an error in the title of 4.2

Corrected.

The title of 4.3 is the same as the title of 4.4

Corrected.

Reviewer 2 Report

The paper "Candidate Causal Variants at the 8p12 Breast Cancer Risk Locus Regulate DUSP4" by Glubb et al. describes a study on a specific locus previously identified as potentially involved in breast cancer by the same group. The authors tested, by a reporter assay, impact on DUSP4 expression of specific variants prioritized by 3C analysis. 

The paper is very well focused and written. My major concern regards the luciferase assay: the regions considered in this study are 300 kb upstream the promoter of DUSP4 and, as pointed by 3C studies, long range interactions are in place within cells. Also, the assay results are in contradiction with GTEx results (as pointed in the discussion). Can the authors rule out the possibility that the assay is not returning a proper readout of the physiological role of the discussed SNPs given that it is not mimicking the actual mechanism of regulation? I would discuss this together with the 3 given motivation for discordance. 

Also, is the assay sensitive enough? When looking at multi-tissue eQTL for rs7461885 and rs12155535, it is clear that they have a similar effect (for example, the both increase the expression of DUSP4 in putamen and small intestine and reduce the expression in Amygdala and cultured fibroblasts). Looking at the assay it is suggestive that both reduce the expression of DUSP4 although only rs7461885 reaches the statistical significance.

More in general, I believe targeted genome editing of the cell lines with the CCVs would be a more natural experiment. 

Lastly, I haven't found matches with another study on eQTL in breast cancer (10.1016/j.ajhg.2018.03.016 ) which may be included in the reference and discussion.

Author Response

My major concern regards the luciferase assay: the regions considered in this study are 300 kb upstream the promoter of DUSP4 and, as pointed by 3C studies, long range interactions are in place within cells.

The reviewer is correct in suggesting that a limitation of the luciferase assay is that as it is plasmid-based it does not mirror the long-range regulatory interactions that occur in the human genome. Nonetheless, enhancers are understood to maintain their functions in reporter plasmids, despite the artificial genetic environment, and reporter assays are considered the ‘gold standard’ for assessing enhancer activity (Shlyueva et al, 2014. Nat Rev Genet; 15:272-86). Indeed, data suggest that distance is not essential for regulatory interaction in a plasmid-based assay: Tewhey et al estimated a positive predictive value of up to 68% when validating eQTLs using reporter plasmids that incorporate only 150 bp of enhancer sequence and a minimal promoter (Cell; 165:1519-1529).

Also, the assay results are in contradiction with GTEx results (as pointed in the discussion)... Also, is the assay sensitive enough? When looking at multi-tissue eQTL for rs7461885 and rs12155535, it is clear that they have a similar effect (for example, the both increase the expression of DUSP4 in putamen and small intestine and reduce the expression in Amygdala and cultured fibroblasts). Looking at the assay it is suggestive that both reduce the expression of DUSP4 although only rs7461885 reaches the statistical significance.

According to the GTEx portal, the statistically significant effects of rs7461885 and rs12155535 (which are in strong linkage disequilibrium) on DUSP4 expression are observed only in cultured fibroblasts. Although the direction of this effect is in fact consistent with our findings (i.e. the risk allele is associated with decreased DUSP4 expression), we observe no co-localisation between the eQTL and GWAS signals i.e. genetic variation other than that represented by the CCVs is likely to have a causal effect on DUSP4 expression and, thus, the association between the CCVs and expression is probably due to linkage disequilibrium between the CCVs and the causal eQTL variation.

Although the eQTL findings in the relevant GTEx tissue (normal mammary tissue) are null and do not support a functional effect of risk variation, we cannot rule out that in a study with more samples (and hence greater statistical power) an effect may be observed. Rather than the reporter gene assay being not sensitive enough, it is perhaps more likely that these eQTL studies do not have power to detect effects of the magnitude observed in these assays (as discussed in lines 175-177).

Can the authors rule out the possibility that the assay is not returning a proper readout of the physiological role of the discussed SNPs given that it is not mimicking the actual mechanism of regulation? I would discuss this together with the 3 given motivation for discordance.

We cannot rule out the possibility that the assay is not returning a proper readout but, as discussed above, luciferase assays are well established as a technique to infer effects on enhancer/promoter activity. We have added text related to this issue to the Discussion (lines 164-171).

More in general, I believe targeted genome editing of the cell lines with the CCVs would be a more natural experiment.

While it could be informative to engineer alleles of the functional risk variant identified, the generation of isogenic cell lines that differ at a single nucleotide locus is not a trivial undertaking and few such studies exist. Moreover, it is questionable whether an effect of 30-50% (as observed in the reporter assays) would be detected using these cellular models.

Lastly, I haven't found matches with another study on eQTL in breast cancer (10.1016/j.ajhg.2018.03.016) which may be included in the reference and discussion.

The authors thank the reviewer for identifying this study which demonstrates that a minority of breast cancer GWAS risk variation show significant associations with gene expression. The study has been added to the Discussion (lines 177-178).

Round 2

Reviewer 2 Report

The authors addressed all issues with reasonable answers.